



# Glacier elevation and mass changes in Himalayas during 2000-2014

Debmita Bandyopadhyay[1], Gulab Singh[1], Anil V.Kulkarni[2]

[1]Center of Studies in Resources Engineering, Indian Institute of Technology Bombay, 400076, India

[2]Divecha Centre for Climate Change, Indian Institute of Science, Bangalore, 560012, India

## Abstract

Glacier mass balance is a crucial parameter to understand the changes in glaciers. For the Himalayas, it is more complex as glaciers have a heterogeneous pattern of elevation and mass changes. In this study,

mass balance using geodetic method is estimated, for which we utilize SRTM and TanDEM-X global digital elevation models (DEMs) of the year 2000 and 2012-2014 respectively. The unique feature of this study is that the dataset are prepared using repeat bistatic synthetic aperture radar interferometry which has not been used over the rugged Himalayan terrains on such a large-scale. The elevation and mass change measurements cover seven states namely Jammu and Kashmir, Himachal Pradesh,

Uttarakhand, Nepal, Sikkim, Bhutan and Arunachal Pradesh. The mean elevation change is $-0.45 \pm 0.40$ m yr$^{-1}$ and the mass budget is $-11.24 \pm 0.79$ Gt yr$^{-1}$. However, the cumulative mass loss over the observation period of 2000-2014 is $-154.72 \pm 19.04$ Gt which accounts for approximately 5% of the total ice-mass present in the Indian Himalayas. This ice-mass loss contributes to $0.42 \pm 0.05$ mm of sea-level rise. Validation of the mass balance estimate over 20 glaciers for which long-term ground

observations were reported gave a coefficient of correlation of 0.79. These 20 glaciers are spread over the entire region of study. Such information shall be helpful in updating the current sparse database we have for the Himalayan glaciers and act as a piece of reliable information for developing various glacier-climate models in the near future.

**Key Words**: Elevation change, Mass budget, Himalayas, SRTM, TanDEM-X

## Introduction

The Himalayan glaciers form the largest mountain glacier systems of the world and also the 'Water Tower of Asia'(Bolch et al., 2012). For India, these glaciers act as a perennial source of water in summer dry months, feeding water into the three major basins of the country namely Indus, Ganges and the Brahmaputra. The glacier melt-water in downstream is utilized for irrigation, hydropower generation

and drinking purpose. However, these glaciers are rapidly losing ice-mass owing to the warming of the climate (Chaturvedi et al., 2014; Immerzeel et al., 2012; Tawde et al., 2017). Even though Karakoram is reported to have positive mass change (Gardelle et al., 2012; Vijay and Braun, 2018), an overall loss in ice-mass for the Himalayas is 443 ±136 Gt of glacier mass in the last 3-4 decades (Kulkarni and Karyakarte, 2014), which certainly influence the water sustainability from these glaciers.



To assess the mass changes in the glaciers, the conventional glaciological (ground-based) method is most sought after. However, the rugged terrains of the Himalayas renders it impossible to continuously monitor such humungous geographical features. The major limitation becomes sparse data-points which are able to locally characterize the area and hence under-represent the changes in the entire Himalayas. With the advent of remote sensing technology, geodetic method has been used as a useful alternate to the ground based measurements. Previous studies in Himalayas have been carried out using either laser altimetry (Kääb et al., 2012) or digital elevation models (DEMs) generated from optical stereo-pairs and Synthetic Aperture Radar (SAR) using interferometry (Agarwal et al., 2017; Berthier et al., 2007; Bolch et al., 2017; Gardelle et al., 2012, 2013). With the recent release of TanDEM-X/TerraSAR-X (since 2010), the geodetic method has been implemented in selective regions of Karakoram-Himalayas using solely SAR images (SRTM and TanDEM-X dataset) (Bandyopadhyay et al., 2018; Vijay and Braun, 2016, 2018). The major advantage of these dataset is the weather independent coverage, as in tropical regions, cloud cover limits visibility to optical images especially in the end of ablation season (September end). Further, this method is one of the simplest and quickest ways by which we can identify the regions that need detailed investigation, especially in remote areas like that of the Himalayan glaciers.

In this study, we utilize freely expended SRTM DEM of the year 2000, and TanDEM-X DEM, which was recently released in September 2018, having scenes collected between 2011 and January 2015. The potential of the new TanDEM-X global DEM has only been tested over the glacierized regions of South-America (Braun et al., 2019) with major contributions from ice-fields and not rugged terrains like the mountain glaciers of Himalayas. This study using bistatic-SRTM and TanDEM-X DEM, would provide an improved insight into the complex behavior of the Himalayas.

**Study Area**

Glacier elevation and mass changes have been estimated over seven Himalayan states of India (Fig. 1). The entire stretch is divided in four regions namely Karakoram (part Jammu and Kashmir (J&K)), western (part of J&K and entire Himachal Pradesh), central (includes Uttarakhand and Nepal) and the eastern Himalayas, which encompass Sikkim, Bhutan and Arunachal Pradesh. Glaciers in these states cover an area of over 40,000 sq. km.

In the east (west Nepal onwards) the glaciers are mainly influenced by the Indian summer monsoon i.e. the accumulation and ablation occurs in the summer season (Fujita, 2008; Gardelle et al., 2013). Whereas, the glaciers in the west mainly Jammu-Kashmir region are affected more by the westerlies i.e. they are winter accumulation type. This leads to an uneven distribution of glaciers with differing response to the changing climatic conditions (Dobhal et al., 2013). For example, the Karakoram host surge-type glaciers (Vijay and Braun, 2018) as opposed to rapidly retreating glaciers in Lahaul-Spiti (Gardelle et al., 2013; Vijay and Braun, 2016). Himachal Pradesh and Uttarakhand lie in the transition zone wherein the glaciers are affected by both summer and winter accumulation. Hence,



a deeper speculation into the inhomogeneous spatial distribution of the change in ice-thickness is performed in this study.

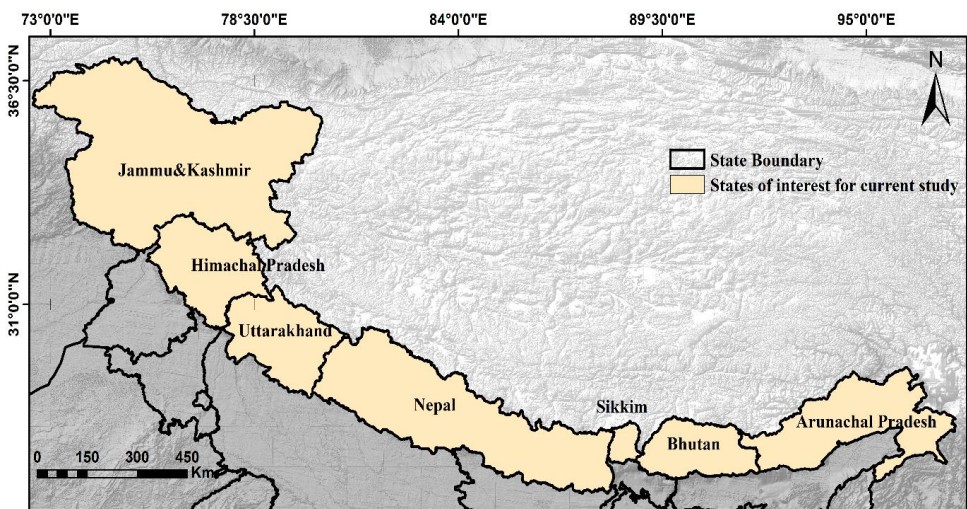

Fig. 1: Study region comprising of Jammu and Kashmir, Himachal Pradesh, Uttarakhand, Nepal, Sikkim, Bhutan and Arunachal Pradesh are highlighted in yellow.

## 1. Dataset and methodology

### 1.1. Database generation

The SRTM DEM is provided by NASA and TanDEM-X DEM by the TerraSAR-X add on for Digital Elevation Measurement mission (TanDEM-X) which is jointly operated by DLR and Astrium Defense and Space. Since, the TanDEM-X coverage is till January 2015, the end date for all our elevation change calculations have considered to be 2014. It must be noted that the calculation has been performed till 2012, as the first acquisitions for global DEM generation was for 2012. Subsequent year DEM generation were done only for correcting the artefacts in DEMs at higher elevations (Personal communication with Papathanassiou, Head of the Information Retrieval Group, DLR on Feb 01 2019). This dataset generation took additional effort in terms of more acquisitions from opposite looking directions to fill in the voids hat were created due to the looking geometry of the satellite. Thus, the DEM TanDEM-X is expected to provide better terrain information than even the 11-day mission of SRTM radar images.

For global coverage, since SRTM-C and TanDEM-X is utilized, the SRTM-X DEM is also used (acquired data in the year 2000) to understand the difference in penetration of C- and X- band over the glaciated and non-glaciated regions. Further, the glacier boundary from Randolph Glacier Inventory (RGI) 6.0 (RGI Consortium, 2017) to extract the mean elevation change over the seven states is



considered for this study. However, the glacier boundaries in this inventory have been updated in the time period of 2003-2009 which is modified using Landsat images from the year 2000.

### 1.2. Bias correction and coregistration

SRTM-X even though sparse in its coverage as compared to SRTM-C band data, the bias correction

has been applied for both on glacier and off-glacier area to account for the penetration bias X- and C-band. It is observed that the penetration difference for glaciated terrains in each state varies and hence the mean penetration bias for each state are as given in Table 1.

Table 1 Mean penetration difference of SRTM-C and SRTM-X over different regions of Indian

Himalayan glaciers

| Region/State | This study | Gardelle et al. (2013) | Kääb et al. (2012) |
|---|---|---|---|
| Jammu and Kashmir/**Karakoram** | 1.34 | **3.4** | 1.4 |
| Himachal Pradesh | 1.79 | - | |
| Uttarakhand | 3.50 | - | 1.5 |
| Nepal/ **Everest** /East-West Nepal/ | 0.75 | **1.4** | |
| Sikkim | 3.01 | - | - |
| Bhutan | 2.40 | 2.4 | 2.5 |
| Arunachal Pradesh | 1.35 | - | - |

Coregistration of the two DEMs is performed to minimize the offset in the horizontal and vertical direction. Since, the DEMs have been acquired in different time periods, by different satellites, in different bandwidths and in different datum, a horizontal and a vertical offset is observed even though

the acquisition is on the same terrain. Nuth and Kääb in 2011 (Nuth and Kaab, 2011) showed that the elevation differences are larger at steeper slopes. Further, the shift can be modelled using a cosine function which is finally normalized to minimize the vertical bias. An iterative method is used till the magnitude of the shift vector is nearly zero. The final equation on which this approach is applied is:

$$\frac{dH}{tan(\alpha)} = a\,cos(b - \psi) + c \tag{1}$$

$$c = \frac{\overline{dH}}{tan(\overline{\alpha})} \tag{2}$$

The three cosine parameters a, b and c are solved using least square minimization and they represent horizontal shift, direction of shift, mean bias divided by mean slope tangent of the terrain respectively. $\alpha$ and $\psi$ are the slope and aspect of the DEM. The threshold for the vertical slope has been kept at $<50°$, assuming 90% area of the glaciers fall below this value. The elevation difference and overall elevation

bias is represented by $dH$ and $\overline{dH}$ respectively.

### 1.3. Elevation change and Mass balance estimation

The DEM differencing was done at a pixel level using the updated glacier boundaries of RGI 6.0 for the study area. For each of the seven states, the difference function (dH/T) i.e. height change in a 14



year period has been generated using 100m bins. The 100m bins have been estimated using the non-void filled SRTM DEM, to give a clearer picture of the elevation change as a function of altitude. This particular methodology acts as base for the hypsometric computations.

The mass budget calculations from the elevation changes consider the density variations of the glaciated terrain to be $850 \pm 60$ kg m$^{-3}$ (Huss, 2013). This density value considers the contribution from all the different features namely snow, firn and ice on a glacier. It might be argued the density variation of these dynamic features might influence our estimated mass budgets. Owing to the sufficiently long observation period, this minor fluctuation could be negated for regions with lower mass balance change rates (Braun et al., 2019).

**1.4. Accuracy assessment**

The mass budget estimate consists of uncertainties mainly from three sources: DEM differencing, glacier outlines and volume to mass conversion factor. For elevation change, the standard method of Normalized Median Absolute Deviation (NMAD) is used. Glacier area uncertainty has been estimated using the following formula (Braun et al., 2019):

$$\delta A = \frac{R_{P/A}}{R_{P/A_{(Paul\ et\ al)}}} * 0.03 \tag{3}$$

where P/A is the perimeter-area ratio and $R_{P/A_{(Paul\ et\ al)}}$ is a constant value of 5.03 km$^{-1}$ (Paul et al., 2013). The volume to mass conversion considers a fixed value $\pm 60$ kg m$^{-3}$ which adds to the error of the mass budget, hence incorporated in the uncertainty estimate.

**2. Results and discussion**

**2.1. Elevation change**

Our investigation shows that the glaciers in the study area have lost -154.72 $\pm$ 19.04 Gt of ice-mass from 2000-2014. This voluminous amount of loss can be attributed to a large amount of elevation change from Jammu and Kashmir. However, it may seem from Fig. 2 that higher negative elevation change is in the Himachal and Nepal region (more towards the red end of the color bar).



**Fig. 2 : Elevation change for the entire Himalayan range with the red color denoting high negative elevation change and blue color representing positive mass change over the period 2000-2014 (enlarged spatial distribution of elevation change is shown in supplementary Fig. S2-S8)**





The elevation change of all the glaciers have been estimated state-wise (Figs. S2-S8). The trend of elevation change over the entire Himalayas (Fig. 2), shows that in the far west near Karakorum, there is an overall positive mass change which has been reported in previously as well. (Gardelle et al., 2012; Rankl and Braun, 2016; Vijay and Braun, 2018). The spatial distribution of the mean elevation change

shows that a high elevation change is observed in Jammu West as compared to Jammu East. This could be due to the surging glaciers which fall in the Karakoram Range. Further as we move towards the East, the percentage of glacierized area also dramatically decreases, which also has an impact on the elevation change rate of the glacier. The elevation change is more for smaller glaciers, which is clearly reflected in spatial distribution pattern.

A longitudinal profile of the elevation change over entire Himalayas (Fig. 3) shows that most of the elevation varies near zero elevation change in Karakoram whereas for rest of Himalayas elevation change is evenly distributed (except Himachal Pradesh). As observed from the distribution, maximum contribution of a near-zero elevation change is observed in the case of Jammu and Kashmir, owing to the high glacierized area or local climatic conditions.

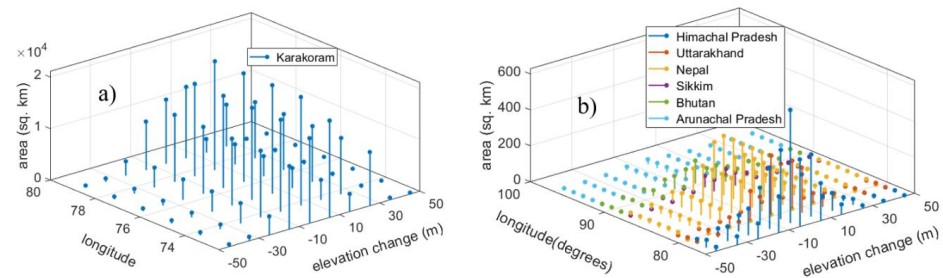

**Fig. 3 : Longitudinal variation of elevation change for the entire Himalayan range where a) represents the Karakoram range and b) represents the rest of six states namely Himachal Pradesh, Uttarakhand, Nepal, Sikkim, Bhutan and Arunachal Pradesh**

The state-wise altitudinal distribution of the elevation change in the seven states namely Jammu and Kashmir, Himachal Pradesh, Uttarakhand, Nepal, Sikkim, Bhutan and Arunachal Pradesh is shown in Fig. 4 (a) to g)). These hypsometry plots give an idea about the equilibrium line altitude (ELA) when the elevation changes from negative to positive. This information is vital for mass balance calculations. In Uttarakhand, Sikkim and Jammu and Kashmir, the hypsometry shows that the elevation change after

approximately 5000m elevation, tend towards a positive value. On the other hand, glaciers in the remaining states except Nepal(wherein the elevation change after ~6800m remains almost zero), always have a negative elevation change regardless of fact that at higher elevation mostly accumulation occurs. This clearly questions the sustainability of these glaciers in the near future. However, it is interesting to note that above 5500m, the overall elevation change in the accumulation zone of the entire Himalayan



glaciers is positive. It is also observed that the spatial variability of Sikkim, Bhutan and Arunachal Pradesh is quite high in the lower altitudes.

  For accuracy assessment, a normalized mean and median absolute deviation (NMAD) for elevation change calculations were performed as shown in the supplement (Fig. S9). The NMAD trend shows that as we move towards higher slope, the absolute deviation about mean/median is significantly higher. However, most of the area of the glaciers are below 60° slope, hence the NMAD values are well within the acceptable range.



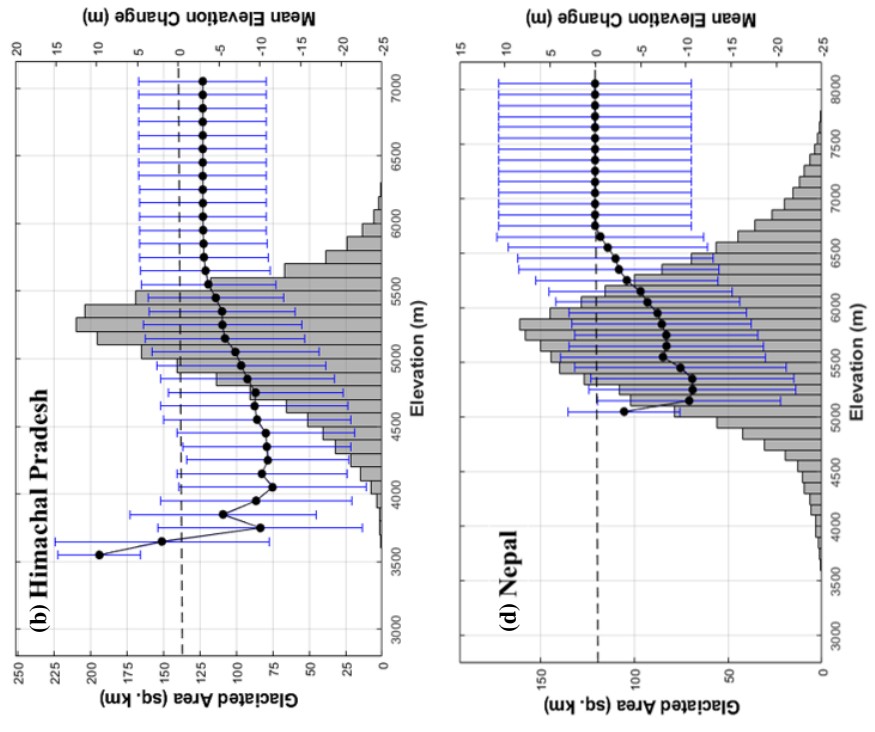

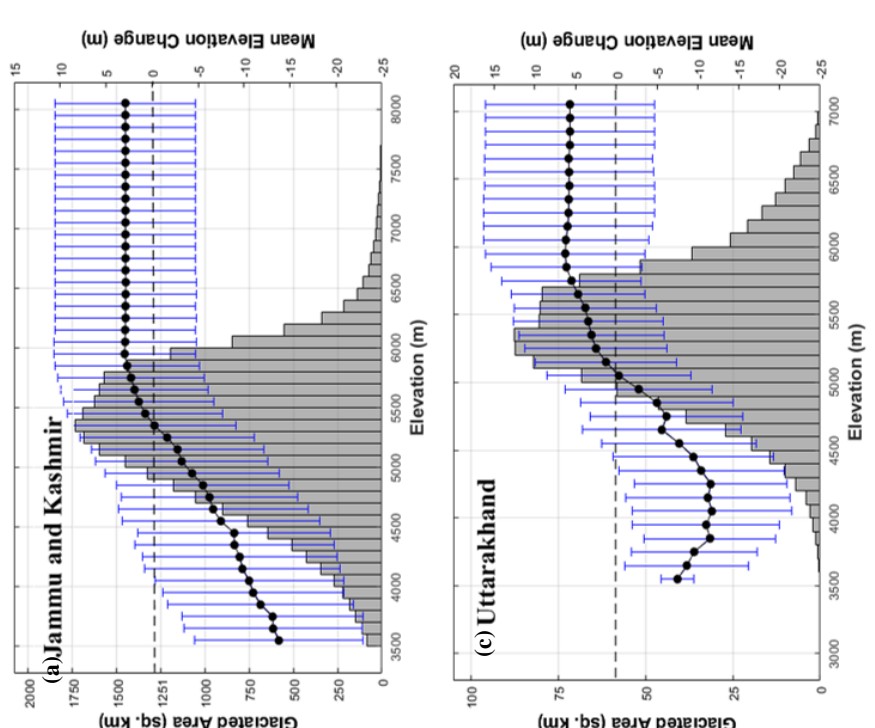





**Fig. 4 :** Hypsometry elevation change plots of a) Jammu and Kashmir, b) Himachal Pradesh, c) Uttarakhand, d) Nepal, e) Sikkim, f) Bhutan and g) Arunachal Pradesh and over entire Himalayas in 100m elevation bins. The blue error bars are representative of the range of mean elevation change at every bin.



### 2.2. Mass Balance

The mass budget has been estimated for the four zones; part of Karakoram, Western, Central and the Eastern Himalayas (Table 2). For reference, the spatial distribution of mass budget four major glaciers namely Siachen (Jammu and Kashmir), Samudra Tapu (Himachal Pradesh), Gangotri (Uttarakhand) and Zemu (Sikkim) is shown (Fig. 5). This pictorial representation gives an idea that the mass budget in general is negative in the ablation zone, however, it may not uniformly follow a gradient as we move down the glacier length from accumulation zone to the ablation zone.

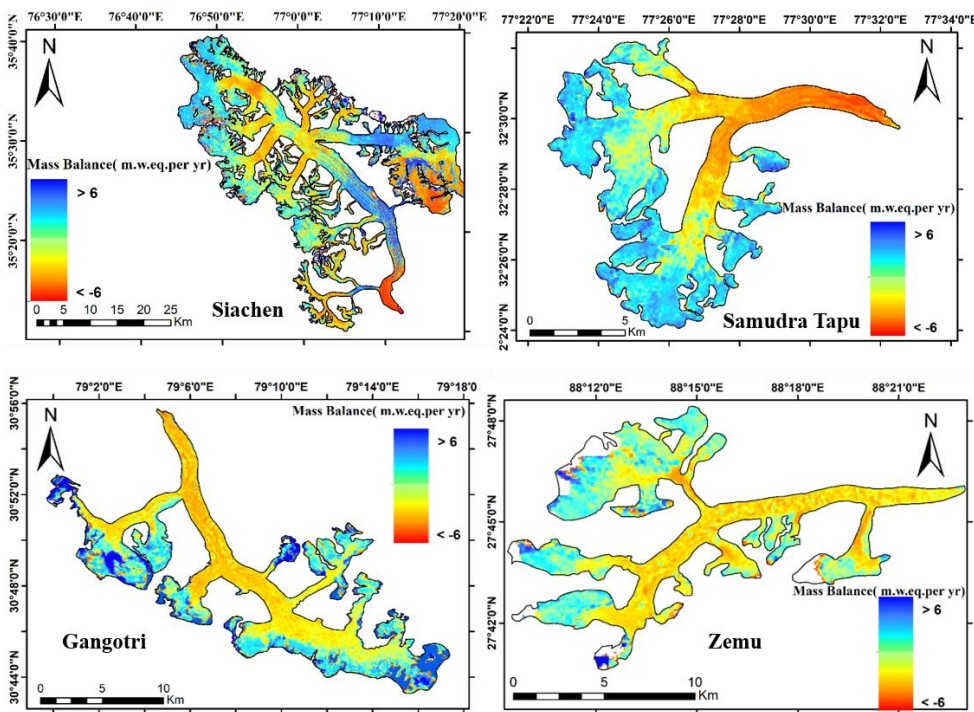

**Fig. 5 : Spatial distribution of mass budget of selective glaciers; Siachen from Jammu and Kashmir, Samudra Tapu from Himachal Pradesh, Gangotri from Uttarakhand and Zemu from Sikkim.**

To validate our measurements over the Himalayas, 20 glaciers spanning over Jammu and Kashmir, Himachal Pradesh, Uttarakhand, Nepal and Bhutan have been chosen (Fig. 6). These are the only glaciers that have reported decadal mass balance. Certain glaciers in Nepal (Sherpa et al., 2017; Wagnon et al., 2013) like Mera (17) and West Changri Nup (19) have been either overestimated or underestimated, largely deviating from the trendline reducing the coefficient of correlation from 0.869 to 0.794 (Fig. 6). The plausible reason for this variable estimates could be due to certain regions of glaciers falling in void region of the DEM and the time period overlap not being the same, which results in either underestimation or overestimation. Even though these 20 glaciers represent only ~0.45% of





the total glaciated area in the Indian Himalayas, regrettably this is the lone ground data available which can be used for authentication of our results.

As seen from Table 2, the last column is the normalized mass budget which is the mass budget divided by the total glaciated area. This calculation is done to highlight the contribution, each state has
towards the total mass budget. Pictorial representation of the same is given in Fig. 7. Here, Jammu and Kashmir, though has the maximum area, has the least mass budget which can be explained from the positive mass change contribution from the Karakoram ranges. This result, is slightly underestimated when compared with the values estimated by Kääb et al.(Kääb et al., 2012), of an elevation change of -0.51 ± 0.06 m yr$^{-1}$ as opposed to our measurements of -0.23 ± 0.01 m yr$^{-1}$. The overall changes in the
Himalayas is found to be -11.05 ± 1.36 Gt yr$^{-1}$ which is comparable to the reported values of -12.8 ± 3.5 Gt yr$^{-1}$ (Kääb et al., 2012). The minor deviation could be attributed to the differing time-scale of observations and study area. Further, the dataset that was used for mass budget in Kääb et al.(Kääb et al., 2012), utilized the sparse data from ICESat and SRTM DEM, whereas in this study we utilize the two dataset with same spatial coverage, resolution as well as mode of acquisition. However, the mass
balance reported for Bhutan by Gardelle et al.(Gardelle et al., 2013) is -0.22 ± 0.13 m w. eq. a$^{-1}$ for 1999-2011 follows the same trend of -0.22 ± 0.01 m w. eq. a$^{-1}$ for 2000-2014 (as calculated from mass budget of -0.26 ± 0.26 Gt yr$^{-1}$).

Table 2 Glacier elevation and mass budget for Indian Himalayas during 2000-2014.

| State | Glacier area (km$^2$) | Mean elevation Change (m /yr ) | Cumulative Mass budget (Gt) | Normalized Mass budget (Gt/yr/km$^2$) |
|---|---|---|---|---|
| Jammu& Kashmir | 25358.4 | -0.23 ± 0.39 | -70.26 ± 3.22 | -1.98*10$^{-4}$ |
| Himachal Pradesh | 3504.7 | -0.59 ± 0.51 | -24.67 ± 1.96 | -5.03*10$^{-4}$ |
| Uttarakhand | 2422.7 | -0.44 ± 0.52 | -15.36 ± 1.92 | -4.43*10$^{-4}$ |
| Nepal | 6637.9 | -0.47 ±0.30 | -36.94 ± 3.33 | -3.98*10$^{-4}$ |
| Sikkim | 494.7 | -0.48 ± 0.33 | -2.86 ± 3.03 | -4.82*10$^{-4}$ |
| Bhutan | 702.2 | -0.26 ±0.26 | -2.19 ± 3.82 | -2.23*10$^{-4}$ |
| Arunachal Pradesh | 350.6 | -0.58 ± 0.57 | -2.42 ± 1.75 | -4.95*10$^{-4}$ |
| Total | 39471.14 | -0.45 ± 0.40 | -154.72 ± 19.04 | -2.68*10$^{-3}$ |

It must be noted that there is a disparity in the mass budget estimated from glaciological method and geodetic method. In fact, it is not expected that the values obtained from the glaciological method would match satellite data derived mass budget. The major reason is the types of processes each of the



method consider. In glaciological method, the surface ice-melt is measured on an annual scale using
stake data. On the other hand, geodetic method encompass not only surface but sub-surface melt
alongside basal melt and even glacier bed-deformation (Paterson, 2010). However, the trend of mass
change over the years using satellite remote sensing, is reflective of the changes occurring on ground
5    (Fig. 6).

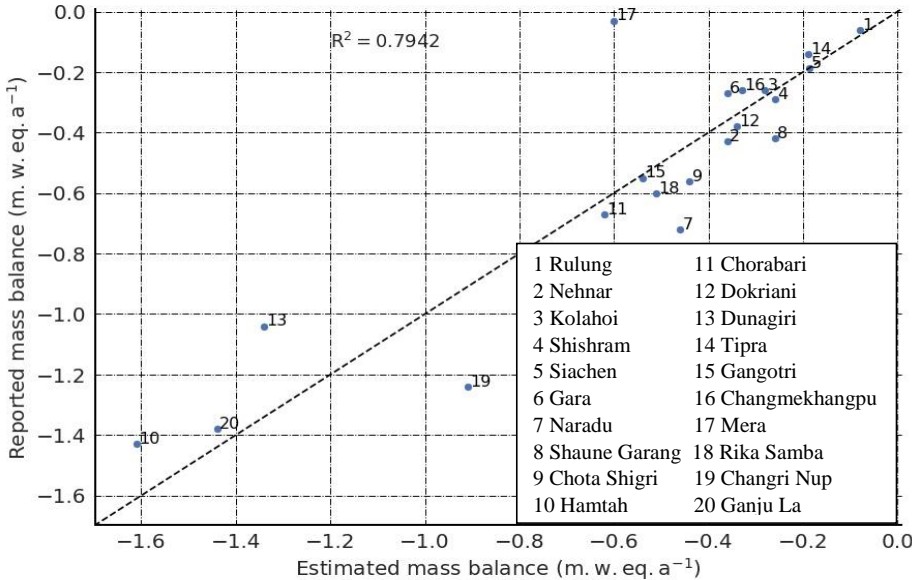

**Fig. 6: Comparison of geodetic mass balance for 20 glaciers with glaciological mass balance (Azam et al., 2018) in Indian Himalayas. The 20 glaciers used for validation of our results were surveyed using glaciological method in Jammu and Kashmir (1-5), Himachal Pradesh (6-10), Uttarakhand (7-15), Sikkim (16), Nepal (17-19) and Bhutan (20).**
10    **Location of these glaciers are shown in Supplementary Fig.s (S1-S7)**

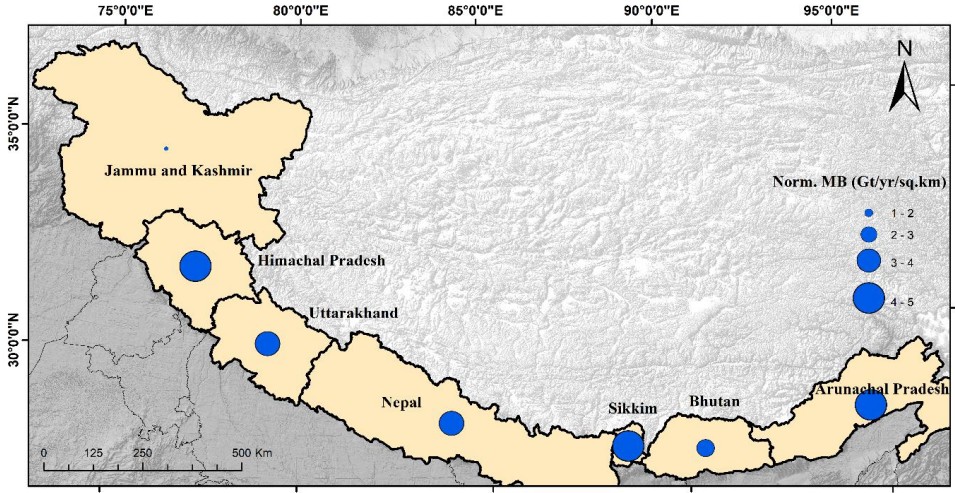

**Fig. 7 : shows the absolute normalized mass budget for each state which is the mass budget per year per sq. km. The size of the blue circles highlight the contribution of mass loss to the total mass loss in Himalayas.**



## Conclusion

The total ice mass loss in 14 years is estimated to be 154.72 ± 19.04 Gt which accounts for more than 0.4 ± 0.05 mm of sea level rise. This is approximately 5% of the total glacier stored ice mass (Kulkarni and Karyakarte, 2014). The total mass budget for 2000-2014 estimated for Himalayas in the seven states is -11.05 ± 1.36 Gt yr$^{-1}$ with a contribution to sea-level rise of 0.03 ± 0.003 mm yr$^{-1}$ due to the glacier melt-water. The mass budget calculated from the far-east of India (Jammu and Kashmir) to the extreme west (Arunachal Pradesh) show no specific trend. However, it is interesting to note that regions with higher glaciated area relatively have a lower rate of ice-mass loss as compared to the states with less glacierized area, anomaly being the state of Bhutan. The geodetic mass balance for Sikkim and Arunachal Pradesh has not been reported before, which certainly will now act as an important addition to the existing sparse information we have about the Himalayas.

This study also highlights the potential of the new bistatic-global TanDEM-X DEM for measuring the elevation and mass changes in steep terrains of the Himalayas. The potential of DEMs solely derived from SAR dataset have been tested for the first time over the rugged terrains wherein data always have a problem of layover and foreshortening. It seems that these dataset, SRTM and TanDEM-X work well as the coefficient of correlation of 0.79 shows good agreement with the ground surveyed glaciers. Our technique of utilizing the new DEMs from the TanDEM-X mission certainly assumes an observation period of 14 years across all states. Nevertheless, the timestamp for each of the acquisitions is different. This might add to the uncertainty of the mass budget, but cannot be directly quantified (Braun et al., 2019). As the scale, of the uncertainty will not be much considering the long observational period, hence we carry forward this information as a legitimate measurement for the glaciers in the Himalayas across the seven states of interest.

A heterogeneous pattern of elevation and mass changes in Himalayas conjecture that an in depth knowledge of the local topography apart from the large-scale climatic parameters like temperature and precipitation need to be considered, especially when glacier modeling is attempted. It also implies that we need more ground observations (meteorological records) on glaciated terrains, to better represent the local atmospheric variables responsible for the varied mass budgets in the Himalayas.

## Acknowledgement

This study is supported by Ministry of Earth Sciences under IMPRINT India (MHRD initiative) with project no. 4096. Authors would like to thank the German aerospace Center (DLR) for kindly providing TanDEM-X data for global DEM. We would also like to thank for Landsat data and SRTM-C data that was freely available from USGS and SRTM-X from DLR.





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
