# Peer review of "Glacier elevation and mass changes in Himalayas during 2000-2014"

_The Cryosphere, 2019_

## Referee Comment (RC1) · Anonymous Referee #1 · 25 Jun 2019

In their TCD manuscript "Glacier elevation and mass changes in Himalayas during 2000–2014" Bandyopadhyay et al. estimated glacier surface elevation changes in the Himalaya mountain range between 2000 and 2014. The authors build their analysis on a digital elevation model acquired in 2000 during the Shuttle Radar Topography Mission and on a more recent TanDEM-X DEM. This study would have had a great impact a couple of years ago, but I have the feeling that in its current form it is lacking novelty in both data processing and relevant glaciological finding. In line with this is the fact that recent and important publications on the topic are not cited. Furthermore, I found the manuscript rather hard to follow (I will not correct for language but the manuscript clearly needs some polishing) and have the feeling that there are several fundamental flaws in the data processing wherefore the manuscript can not be accepted in its

current form. I will give some suggestions in the following but I am not sure if the necessary changes can be made within a review. I am really sorry to not be more positive, but I think the authors need to do a proper literature study first in order to place their results into the current research context.

Introduction

This section needs some language polishing, furthermore some references are wrongly placed and some are missing.

Page 1 Line 29: maybe Pritchard (2019) would be a good reference here?

Page 1 Line 31: what about Brun et al. (2017)? This is one of the most important recent studies but is not cited at all.

Page 2 Lines 8-10: what about Lin et al. (2017)? They are using TanDEM-X SAR data for large parts of the study region.

Page 2 Line 20: to my knowledge Braun et al. (2019) did not rely on the TanDEM-X global DEM but process DEMs by themselves. Please double check.

Study area

I am not sure if the authors should use state boarders to separate their study areas. In order to compare results to other studies (e.g. Kääb et al. 2015, Brun er al. 2017) it would be advisable to use their sub-regions or grid the data. Even though the recent study of Maurer et al., 2019 became available after submitting the initial manuscript I very much like their way of presenting results.

Dataset and methodology

Overall I find this section hard to follow, but I presume the authors rely on the global TanDEM-X DEM rather than processing DEMs by themselves? This might be ok, but need to be done in a proper way. In the following I give several fundamental suggestions which need to be accounted for.

[Figure]

1.) Both the SRTM and TanDEM-X global DEM come with a lot of metadata such as error and coverage maps. For example, it is not sufficient to state that the TanDEM-X DEM is from 2014 as this is simply not true. Instead, the authors need to rely on the exact metadata when calculating yearly elevation changes otherwise the results are biased.

2.) There are many versions of the SRTM DEM available some are void filled and some are not. It is not clear which version and at which grid posting the data were used. The latter also applies for the TanDEM-X DEM. I further recommend to read the study of Mukul et al. (2017) to gain a better understanding of errors in the SRTM dataset over India.

3.) Page 3 Lines 21-23: radar penetration depth is a very important point which is widely discussed in the recent literature. It is not clear how the authors correct for this bias. I strongly suggest to read more recent studies dealing with this topic, focusing explicitly on TanDEM-X data (see for example Dehecq, A. et al., 2016, Vijay, S. et al., 2016, Neelmeijer, J. et al., 2017 Abdel Jaber et al., 2018 and Kääb et al., 2018). This point needs much more consideration. Although SRTM-X and TanDEM-X were acquired at the same wavelength, surface properties could still have been different in both years.

Page 4 Lines 1-2: Did the authors update the dataset by themselves? Not clear. How can the time period 2003-2009 be updated with data from 2000? Please clarify.

Page 5 Lines 1-3: how did the authors account for voids? Not clear but important. Please see also McNabb et al., 2019 on this issue.

Results and Discussion

I will not put too much effort into this section as I presume it will be change quite a bit after revision.

Figure 2: I am sorry, but I can not see much here. . . please use another form of presenting your results, see also my comment on the study area. For inspiration have a look at Brun et al., 2017 or Maurer et al., 2019.

Figure 5: In order to gain a better feeling on the quality of the dataset it would be great to also show off-glacier elevation changes instead of cropping the elevation changes with a glacier mask (the same applies for Figures S2-S8).

Page 11: I very much like the idea to compare the results to in-situ mass balance measurements. However, I have the feeling more effort could be put into this. See also Fischer (2011) on this issue.

Page 11 Line 11: Please compare your results also to the estimates from Brun et al. 2017. Their results are available here: https://doi.pangaea.de/10.1594/PANGAEA.876545

Page 11 Lines 16-18: possibly true but this can be investigated further. Again please see McNabb et al., 2019 concerning the void issue and the TanDEM-X metadata concerning the time issue. Further, penetration bias and density assumption will have an effect and need to be discussed. Maybe this is a little bit beyond the scope of the study but how compare the results of Brun et al. 2017 to these in-situ measurements?

Conclusions

Page 14 Lines 12-15: this is not true. See for example Rankl et al. 2016, Lin et al. 2017 and Neelmeijer et al. 2017.

Page 14 Lines 19-20: I think this can be further quantified by investigating the metadata of the TanDEM-X DEM. As stated above Braun et al. (2019) did not rely on the global TanDEM-X DEM.

Page 14 Line 20: This is not true. If the authors calculate annual elevation changes between 2000 and 2014 but the correct end date is actually 2011 the results are significantly biased.

Additional References:

Abdel Jaber, W., Rott, H., Floricioiu, D., Wuite, J., and Miranda, N. (2018): Heterogeneous spatial and temporal pattern of surface elevation change and mass balance of the Patagonian icefields between 2000 and 2016, The Cryosphere Discuss., https://doi.org/10.5194/tc-2018-258, in review.

Berthier, E., Larsen, C., Durkin, W. J., Willis, M. J., and Pritchard, M. E. (2018): Brief communication: Unabated wastage of the Juneau and Stikine icefields (southeast Alaska) in the early 21st century, The Cryosphere, 12, 1523-1530, https://doi.org/10.5194/tc-12-1523-2018.

Brun F, Berthier E, Wagnon P, Kääb A and Treichler D (2017): A spatially resolved estimate of High Mountain Asia glacier mass balances from 2000–2016 Nat. Geosci. 10 668–73

Dehecq, A., Millan, R., Berthier, E., Gourmelen, N., Trouvé, E. and Vionnet, V. (2016): "Elevation Changes Inferred From TanDEM-X Data Over the Mont-Blanc Area: Impact of the X-Band Interferometric Bias," in IEEE Journal of Selected Topics in Applied Earth Observations and Remote Sensing, vol. 9, no. 8, pp. 3870-3882, doi: 10.1109/JSTARS.2016.2581482

Fischer, A. (2011): Comparison of direct and geodetic mass balances on a multi-annual time scale, The Cryosphere, 5, 107-124, https://doi.org/10.5194/tc-5-107-2011

Kääb A and 18 others (2018) Massive collapse of two glaciers in western Tibet in 2016 after surge-like instability. Nat. Geosci., 11, 114–120 (doi: 10.1038/s41561-017-0039-7)

Li G, Lin H and Ye Q (2018) Heterogeneous decadal glacier downwasting at the Mt. Everest (Qomolangma) from 2000 to similar to 2012 based on multi-baseline bistatic SAR interferometry. Remote Sens. Environ., 206, 336–349 (doi: 10.1016/j.rse.2017.12.032)

Lin H, Li G, Cuo L, Hooper A and Ye Q (2017): A decreasing glacier mass balance gradient from the edge of the Upper Tarim Basin to the Karakoram during 2000–2014. Sci. Rep., 7(1), 6712 (doi:10.1038/s41598-017-07133-8)

Maurer, J. M., Schaefer, J. M., Rupper, S. and Corley, A. (2019): Acceleration of ice loss across the Himalayas over the past 40 years. Sci. Adv. 5, eaav7266.

McNabb, R., Nuth, C., Kääb, A., and Girod, L. (2019): Sensitivity of glacier volume change estimation to DEM void interpolation, The Cryosphere, 13, 895-910, https://doi.org/10.5194/tc-13-895-2019.

Mukul, M.; Srivastava, V.; Jade, S.; Mukul, M. Uncertainties in the Shuttle Radar Topography Mission (SRTM) Heights: Insights from the Indian Himalaya and Peninsula. Sci. Rep. 2017, 7, 41672.

Neelmeijer, J., Motagh, M., Bookhagen, B. (2017): High-resolution digital elevation models from single-pass TanDEM-X interferometry over mountainuous region: a case study of Inylchek Glacier, Central Asia. ISPRS J. Photogramm. Remote Sens. 130, 108–121. http://dx.doi.org/10.1016/j.isprsjprs.2017.05.011.

Pritchard, Hamish. (2019): Asia's shrinking glaciers protect large populations from drought stress. Nature, 569. 649-654. 10.1038/s41586-019-1240-1

---

## Referee Comment (RC2) · Anonymous Referee #2 · 1 Jul 2019

General comments

In this manuscript, the authors present the glacier elevation and mass changes in the Himalayas covering a period between 2000 and 2014. They derived elevation changes from digital elevation models such as SRTM-C band and TanDEM-X global DEM. The Himalayas, of course, is an area of great interest for many communities due to the relevance and role of glaciers as water supplies. In that sense, additional information and results are always very welcome, and we acknowledge the authors effort.

Overall, the study is presented in a very simple way (e.g. methods) that do not completely support the content in the abstract. I can understand the short description of the methodology due to the authors using two DEMs with global coverage. However, the present manuscript suffers from various conceptual limitations and methodological

inconsistencies that do not provide assurance of the quality of the results. Therefore, it is my recommendation that this paper is not suitable for publication in its current form. The reasons are the following:

1) One of the key statements or motivation from the authors in this manuscript is that TanDEM-X "Global" DEM has been recently used to calculate glacier elevation and mass changes in South America by Braun et al., (2019). They (Braun et al 2019) did not use the TanDEM-X "Global" DEM. We have to make the difference here. Braun and colleagues (2019) processed hundreds of raw radar images (InSAR) to generate their own TanDEM-X DEM, concentrated in the ablation period. There are also many others studies dealing with TanDEM-X processing that carried out similar procedures (e.g. Necklel et al., 2013; Rankl et al., 2016; Dehecq et al., 2016; Neelmeijer et al., 2017; Vijay et al., 2017; Malz et al., 2018; Abdel Jaber, 2018; Rott et al., 2018). Neelmeijer et al., (2017), provide a clear overview of the processing chain of TanDEM-X in figure 3 or Dehecq et al, (2016) in figure 2. On the other hand, the TanDEM-X Global DEM is an effort from DLR (German Space Agency) to cover the entire globe with low and high-resolution DEM (12 to 90 m) with thousands of intermediate DEMs to generate this large globe DEM mosaic. Unlike of SRTM DEM (February of 2000), the dates for the composition TanDEM-X "GLOBAL" DEM mosaic is unknown or at least not easily found. Which means for the global DEM, we can find different intermediate DEM seasons. This led to one of the major uncertainties: what are the dates of the all intermediate DEM in Himalayas? However, although the TanDEM-X "Global" DEM is a very sophisticated DEM that provides a useful topography for many other fields, for glacier elevation changes calculations may lead to more uncertainties. Hence, I leave this point to the discretion of the editor.

2) I also agree with the reviewer #1 that the present manuscript is missing several previous studies either for the methodology description (e.g. Rankl et al., 2016; Dehecq et al., 2016; Neelmeijer et al., 2017) or for the study area (e.g. King et al., 2017; Brun et al., 2017). The most critical study is from Brun et al., (2017). Brun and colleagues

(2017) calculated glacier elevation changes very close to the period of the present study (2000-2016). This is a pity, since the authors could compare their results using the same catchment/basin subdivisions and use the opportunity to compare it.

3) The description of the methods and uncertainties section, as I stated before, are not specific enough. I have a lot of doubts about the methodology and the interpretation of results that the authors present in this manuscript, along with the important confusion of the used materials. I will also give you some suggestions, however, substantial work will be needed.

1 Methods section

P2 L17 -> What do you define as rugged terrain? In some areas of the Andes is reaching almost 7000 m a.s.l.

P3 L10-15 -> it is very simplistic to state as 2014, where not further information is provided. As I mentioned above, please try to provide a realistic date to trace the results. I agree with reviewer #1 that the results are biased.

P4 L3-6 -> how was the radar signal penetration considered? It is not precisely described. I am not fully convinced with the values showed by the authors. Other examples dealing with X and C band penetration showed much more radar signal penetration (e.g Dehecq et al., (2016); Neelmeijer et al., (2017); Vijay et al., (2017) (see my suggestion below).

P5 L1-3 -> No detailed information about the hypsometry computation and error assessment.

P5 L10-19 -> There is information is missing in this section (a) no description for the NMAD method (see Höhle and Höhle, 2009). (b) What equation contains the total uncertainty of your study? (e.g. Vijay et al., 2017; Braun et al., 2019).

Despite the methodology is not precise. From what I have seen, the manuscript does not reflect the state of art regarding TanDEM-X assessment (e.g. Neelmeijer et al.,

2017; Vijay et al., 2017; Malz et al., 2018; Abdel Jaber, 2018; Rott et al., 2018; Braun et al., 2019). Therefore, I suggest a complete re-analysis /re-organization, including in your analysis:

a. Uncertainty from the volume to mass conversion. Please use density scenarios (due to multi-seasonality/dates).

b. Uncertainty from radar signal penetration. Vijay and Braun (2016) showed that there is a strong altitude dependency of the radar signal penetration bias. They observed a range from 0.84 m (5000 m a.s.l.) to 3.64 m (5800 m a.s.l.). Since the date of season/year of TanDEM-X "Global" DEM is unknown I would use the worst scenario of radar signal penetration. A good example is in Braun et al., (2019), although they calculated glacier mass changes in the ablation period, they applied a radar signal penetration from 0 to 5 m from Equilibrium Line Altitude (ELA), considering negligible below to ELA. Other example is given by Neelmeijer et al., (2017). You should consider similar procedures.

c. Uncertainty by the hypsometry (please see Berthier et al., 2016; Brun et al., 2017; Vijay et al., 2016; Braun et al., 2019)

d. Error from the DEM differencing (please see Berthier et al., 2016; Vijay et al., 2016; Brun et al., 2017; Braun et al., 2019)

e. Error from the glacier outlines (please see Berthier et al., 2016; Brun et al., 2017; Braun et al., 2019)

f. Uncertainty by the dates. This point requires investigation/analysis. It would be good if the authors can get some originals intermediate DEMs of TanDEM-X "GLOBAL" to check some dates.

2 Results and discussion section

From the facts that I mentioned above, this section does not provide reliable results. Furthermore, the figures do not help too much. In the following I give you some suggestions that could improve this section:

-First, I would separate results and discussion, since there are some topics you have to properly discuss. e.g. A section on your error assessment with the proposed accuracy assessment methodology. Comparison with other studies and comparison with your glacier mass balance dataset (glaciological method).

-I also suggest you use the catchment/sub-division used by Brun et al., 2017 or Deheq et al., 2018 in order to have comparable numbers in your results and the discussion.

-In the last few weeks a couple of papers came out with new insight in this region. It would be good to include it (see: Zemp et al., 2019, Wouters et al., 2019; Maurer et al., 2019).

Figures

P6 Figure 5 -> I agree with reviewer #1. For such a big area I am not sure if this is a representative figure. Please also check Menounos et al., (2019) or Kääb et al., (2012) there are some useful figures that you could apply.

P7 Figure 2 -> with a better quality of figure 1 you can remove figure 2.

P9 and P10 Figure 4 -> some figures are not well represented in the main text. Principally in the discussion. For example, figure 4 d, f, and g. These hypsometric plots present patterns that should be discussed.

3 Reference

Abdel Jaber, W., Rott, H., Floricioiu, D., Wuite, J., and Miranda, N.: Heterogeneous spatial and temporal pattern of surface elevation change and mass balance of the Patagonian icefields between 2000 and 2016, The Cryosphere Discuss., https://doi.org/10.5194/tc-2018-258, in review, 2018.

Berthier, E., Cabot, V., Vincent, C., and Six, D.: Decadal region-wide, and glacier-wide mass balances derived from multi-temporal ASTER satellite digital elevation models. Validation over the Mont-Blanc area, Front. Earth Sci., 4, 63, https://doi.org/10.3389/feart.2016.00063, 2016. 

Braun, M. H., Malz, P., Sommer, C., Farías-Barahona, D., Sauter, T., Casassa, G., et al. (2019). Constraining glacier elevation and mass changes in South America. Nat. Clim. Change 9, 130–136. doi: 10.1038/s41558-018-0375-710.1038/s41558-018-0375-7

Brun, F., Berthier, E., Wagnon, P., Kääb, A., and Treichler, D.: A spatially resolved estimate of High Mountain Asia glacier mass balances from 2000 to 2016, Nat. Geosci., 10, 668–673, https://doi.org/10.1038/ngeo2999, 2017

Dehecq, A., Millan, R., Berthier, E., Gourmelen, N., Trouvé, E. and Vionnet, V. (2016): "Elevation Changes Inferred From TanDEM-X Data Over the Mont-Blanc Area: Impact of the X-Band Interferometric Bias," in IEEE Journal of Selected Topics in Applied Earth Observations and Remote Sensing, vol. 9, no. 8, pp. 3870-3882, doi: 10.1109/JSTARS.2016.2581482

Dehecq A and 9 others (2019) Twenty-first century glacier slowdown driven by mass loss in High Mountain Asia. Nat. Geosci., 12(1), 22–27 (doi: 10.1038/s41561-018-0271-9)

King, O., Quincey, D. J., Carrivick, J. L., and Rowan, A. V.: Spatial variability in mass loss of glaciers in the Everest region, central Himalayas, between 2000 and 2015, The Cryosphere, 11, 407-426, https://doi.org/10.5194/tc-11-407-2017, 2017.

Höhle, J., Höhle, M., 2009. Accuracy assessment of digital elevation models by means of robust statistical methods. ISPRS J. Photogramm. Remote Sens. 64, 398–406. http://dx.doi.org/10.1016/j.isprsjprs.2009.02.003.

Kääb, A., Berthier, E., Nuth, C., Gardelle, J., Arnaud, Y., 2012. Contrasting patterns of early twenty-first-century glacier mass change in the Himalayas. Nature 488, 495–498. http://dx.doi.org/10.1038/nature11324.

Neckel, N., Braun, A., KropácËĞek, J., Hochschild, V., 2013. Recent mass balance of

the Purogangri Ice Cap, central Tibetan Plateau, by means of differential X-band SAR interferometry. The Cryosphere 7, 1623–1633. http://dx.doi.org/10.5194/tc-7- 1623-2013

Malz, P., Meier, W., Casassa, G., Jaña, R., Skvarca, P., Braun, M.H. Elevation and Mass Changes of the Southern Patagonia Icefield Derived from TanDEM-X and SRTM Data. Remote Sens. 2018, 10, 188.

Maurer, J. M., Schaefer, J. M., Rupper, S. and Corley, A. (2019): Acceleration of ice loss across the Himalayas over the past 40 years. Sci. Adv. 5, eaav7266.

Menounos, B., Hugonnet, R., Shean, D., Gardner, A., Howat, I., Berthier, E., et al. (2019). Heterogeneous changes in western North American glaciers linked to decadal variability in zonal wind strength. Geophysical Research Letters, 46, 200–209. https://doi.org/10.1029/ 2018GL080942

Rott, H., Abdel Jaber, W., Wuite, J., Scheiblauer, S., Floricioiu, D., van Wessem, J. M., Nagler, T., Miranda, N., and van den Broeke, M. R.: Changing pattern of ice flow and mass balance for glaciers discharging into the Larsen A and B embayments, Antarctic Peninsula, 2011 to 2016, The Cryosphere, 12, 1273-1291, https://doi.org/10.5194/tc-12-1273-2018, 2018.

Vijay, S. and Braun, M.H. (2016). Elevation Change Rates of Glaciers in the Lahaul-Spiti (Western Himalaya, India) during 2000–2012 and 2012–2013. Remote Sens. 8, 1038. https://doi.org/10.3390/rs8121038

Wouters. B., Gardner, A.S., and Moholdt, G. (2019). Global glacier mass loss during the GRACE Satellite Mission (2002-2016). Front. Earth Sci. 7:96. doi: 10.3389/feart.2019.0009

Zemp, M., Huss, M., Thibert, E., Eckert, N., McNabb, R., Huber, J., Barandun, M., Machguth, H., Nussbaumer, S.U., Gärtner-Roer, I., Thomson, L., Paul, F., Maussion, F., Kutuzov, S., and Cogley, J.G. (2019): Global glacier mass changes and

their contributions to sea-level rise from 1961 to 2016. Nature 568 (7752), 382-386. 10.1038/s41586-019-1071-0

---

## Author Comment (AC1) · 24 Jul 2019

1.Introduction

Comment: Page 1 Line 29: maybe Pritchard (2019) would be a good reference here? #Reply: Reference seems to be published at a date after the submission of the manuscript and not relevant to the statement suggested for.

Comment: Page 1 Line 31: what about Brun et al. (2017)? This is one of the most important recent studies but is not cited at all. #Reply: We thank the reviewer for pointing out the reference and the authors shall incorporate this in the manuscript.

Comment: Page 2 Lines 8-10: what about Lin et al. (2017)? They are using TanDEM-X SAR data for large parts of the study region. #Reply: The authors are trying to

highlight the usage of the freely disseminated global TanDEM-X data. Further, the reference cited by the reviewer used 39 pairs of TanDEM-X SAR data for interferogram generation, which is not the same dataset for the current study.

Comment: Page 2 Line 20: to my knowledge Braun et al. (2019) did not rely on the TanDEM-X global DEM but process DEMs by themselves. Please double check. #Reply: The suggestion of the reviewer is duly considered and the necessary changes will be made.

2.Study area

Comment: I am not sure if the authors should use state boarders to separate their study areas. In order to compare results to other studies (e.g. Kääb et al. 2015, Brun er al. 2017) it would be advisable to use their sub-regions or grid the data. Even though the recent study of Maurer et al., 2019 became available after submitting the initial manuscript I very much like their way of presenting results. #Reply: The idea was to highlight how each state is performing in terms of mass loss alongside easy comparison with Kääb et al.,2012 as they also performed certain regional studies in a state-wise manner (Table 1, page 497). Therefore, even though the reviewer suggests a different demarcation of region of study, we would like to keep the current state boundaries to highlight the issue, which holds utmost importance in an agrarian economy like India.

3. Dataset and methodology

Overall I find this section hard to follow, but I presume the authors rely on the global TanDEM-X DEM rather than processing DEMs by themselves? This might be ok, but need to be done in a proper way. In the following I give several fundamental suggestion which need to be accounted for. Both the SRTM and TanDEM-X global DEM come with a lot of metadata such as error and coverage maps. For example, it is not sufficient to state that the TanDEM-X DEM is from 2014 as this is simply not true. Instead, the authors need to rely on the exact metadata when calculating yearly elevation changes

otherwise the results are biased.

a) Comment: Both the SRTM and TanDEM-X global DEM come with a lot of metadata such as error and coverage maps. For example, it is not sufficient to state that the TanDEM-X DEM is from 2014 as this is simply not true. Instead, the authors need to rely on the exact metadata when calculating yearly elevation changes otherwise the results are biased. #Reply: The exact date for the TanDEM-X DEM is not possible to state. In fact one DEM tile has multiple acquisitions with significant contribution from each year between 2011 and 2014. For example, in N32E79_DEM, there are 20 acquisitions from the year 2011, 15 acquisitions from the year 2012, 30 acquisitions in 2013 and 26 acquisitions in 2014. The baselines are varying between 95 to 200 and the scenes such that they cover the entire area each year, updating the DEM height information for entire area every year. So each yea, when the acquisition is made for months varying from February to December, the information in the DEM is updated, a mosaic of information provided in finished product. Therefore, it if difficult to provide a specific date. However, to make sure that the authors are not providing wrong information to the readers of this prestigious journal, we made sure to compare the results with reported literature, finding a good correlation of 0.79. Hence, proving that the result presented in this manuscript are not speculative but rigorously evaluated.

b) Comment: There are many versions of the SRTM DEM available some are void filled and some are not. It is not clear which version and at which grid posting the data were used. The latter also applies for the TanDEM-X DEM. I further recommend to read the study of Mukul et al. (2017) to gain a better understanding of errors in the SRTM dataset over India. #Reply: The TanDEM-X global DEM and version 2 of the SRTM DEM with 90m posting were utilized, which shall be clarified in the manuscript.

c) Comment: Page 3 Lines 21-23: radar penetration depth is a very important point which is widely discussed in the recent literature. It is not clear how the authors correct for this bias. I strongly suggest to read more recent studies dealing with this topic, focusing explicitly on TanDEM-X data (see for example Dehecq, A. et al., 2016, Vijay, S.

et al., 2016, Neelmeijer, J. et al., 2017 Abdel Jaber et al., 2018 and Kääb et al., 2018). This point needs much more consideration. Although SRTM-X and TanDEM-X were acquired at the same wavelength, surface properties could still have been different in both years. #Reply: The radar penetration has been well addressed in the manuscript (section 1.2). However, the authors understand the concern and would like to bring to the notice of the reviewer that penetration of X-band is hardly 40cm considering the wetness (0.5% by vol.) snowpack covered glacier area for different seasons(Manickam et al., 2017; Singh et al., 2017). Himalayan glaciers have snowpack that have moisture throughout February- September [a few centimeter penetration though the snowpack at X-band]. Hence, this X-band bias shall not effect much after inclusion of DEM and penetration bias, which already has been performed in the current study.

d) Comment: Page 4 Lines 1-2: Did the authors update the dataset by themselves? Not clear. How can the time period 2003-2009 be updated with data from 2000? Please clarify. #Reply: The RGI inventory shape files for glaciers have been modified according to the period of study i.e. the year 2000. This clarification shall be made as per reviewer suggestions.

e) Comment: Page 5 Lines 1-3: how did the authors account for voids? Not clear but important. Please see also McNabb et al., 2019 on this issue. #Reply: If >50% of the glacier is not covered in the dataset used, we discard the area and not use void-filling as the results would be significantly biased.

4.Result and discussion

a) Comment: Figure 2: I am sorry, but I cannot see much here: please use another form of presenting your results, see also my comment on the study area. For inspiration have a look at Brun et al., 2017 or Maurer et al., 2019 #Reply: Gridded data as referred in Brun et al., have been performed over a 111x111 sq. km grid. The trend we show here is on 90m X 90m scale. Hence the disparity. Representing in suggested format might result in under-representing the regions we want to highlight.

b) Comment: Figure 5: In order to gain a better feeling on the quality of the dataset it would be great to also show off-glacier elevation changes instead of cropping the elevation changes with a glacier mask (the same applies for Figures S2-S8). #Reply: Off-glacier elevation changes have been considered in the DEM bias corrections. Furthermore, the focus of the study is only the glaciated terrain, hence extracted in figures S2-S8. This representation is widely accepted in previous studies as well (Lin et al., 2017, Vijay and Braun, 2016 and 2018).

c) Comment: Page 11 Line 11: Please compare your results also to the estimates from Brun et al. 2017. Their results are available here: https://doi.pangaea.de/10.1594/PANGAEA.876545 Reply: The results in Brun et al, 2017 show basin-wise results which are more pertinent if HMA studies are performed. Cross-check with only one state i.e. Bhutan when performed differs from the result reported by 0.02 m.w.eq. per year.

d) Comment: Page 11 Lines 16-18: possibly true but this can be investigated further. Again please see McNabb et al., 2019 concerning the void issue and the TanDEM-X metadata concerning the time issue. Further, penetration bias and density assumption will have an effect and need to be discussed. Maybe this is a little bit beyond the scope of the study but how compare the results of Brun et al. 2017 to these in-situ measurements? #Reply: Brun et al, 2017 mentions mass changes for basins which are beyond the region of interest in current study and thus it is not advisable to use for comparison of results. Further, the density assumption is well-established (Huss (2013)) and used by Gardelle et al. (2011),Vijay and Braun (2016).

5.Conclusion

a) Comment: Page 14 Lines 12-15: this is not true. See for example Rankl et al. 2016, Lin et al.2017 and Neelmeijer et al. 2017. #Reply: The authors thank the reviewer to point this out with reference. Necessary changes will be made in the revised manuscript.

b) Comment: Page 14 Lines 19-20: I think this can be further quantified by investigating the metadata of the TanDEM-X DEM. As stated above Braun et al. (2019) did not rely on the global TanDEM-X DEM. #Reply: This suggestion has been answered in Comment 1 (in Dataset and methodology).

c) Comment: Page 14 Line 20: This is not true. If the authors calculate annual elevation changes between 2000 and 2014 but the correct end date is actually 2011 the results are significantly biased. #Reply: The acquisitions are till Jan 2015 (1-2 acquisitions per DEM as opposed to 20-30 acquisitions of previous year) with the complete year of 2014 covered.

References

Braun, A. M. H., Malz, P., Sommer, C. and Barahona, D. F.: Constraining glacier elevation and mass changes in South America, Nature Climate Change letters, 9(region 03), 130–136, doi:10.1038/s41558-018-0375-7, 2019.

Gardelle, J., Berthier, E., Arnaud, Y. and Kääb, A.: Region-wide glacier mass balances over the Pamir-Karakoram-Himalaya during 1999–2011, Cryosphere, 7(4), 1263–1286, doi:10.5194/tc-7-1263-2013, 2013.

Kääb, A., Berthier, E., Nuth, C., Gardelle, J. and Arnaud, Y.: Contrasting patterns of early twenty-first-century glacier mass change in the Himalayas, Nature, 488(7412), 495–498, doi:10.1038/nature11324, 2012.

Manickam, S., Bhattacharya, A., Singh, G. and Yamaguchi, Y.: Estimation of Snow Surface Dielectric Constant from Polarimetric SAR Data, IEEE Journal of Selected Topics in Applied Earth Observations and Remote Sensing, 10(1), 211–218, doi:10.1109/JSTARS.2016.2588531, 2017.

Singh, G., Verma, A., Kumar, S., Snehmani, Ganju, A., Yamaguchi, Y. and Kulkarni, A. V.: Snowpack Density Retrieval Using Fully Polarimetric TerraSAR-X Data in the Himalayas, IEEE Transactions on Geoscience and Remote Sensing, 55(11), 6320–

6329, doi:10.1109/TGRS.2017.2725979, 2017.

Vijay, S. and Braun, M.: Elevation change rates of glaciers in the Lahaul-Spiti (Western Himalaya, India) during 2000-2012 and 2012-2013, Remote Sensing, 8(12), 1–16, doi:10.3390/rs8121038, 2016.

Vijay, S. and Braun, M.: Early 21st century spatially detailed elevation changes of Jammu and Kashmir glaciers (Karakoram–Himalaya), Global and Planetary Change, 165(April), 137–146, doi:10.1016/j.gloplacha.2018.03.014, 2018.

---

## Author Comment (AC2) · 24 Jul 2019

1.Comment: One of the key statements or motivation from the authors in this manuscript is that TanDEM-X "Global" DEM has been recently used to calculate glacier elevation and mass changes in South America by Braun et al., (2019). They (Braun et al 2019) did not use the TanDEM-X "Global" DEM. We have to make the difference here. Braun and colleagues (2019) processed hundreds of raw radar images (InSAR) to generate their own TanDEM-X DEM, concentrated in the ablation period. There are also many oth- ers studies dealing with TanDEM-X processing that carried out similar procedures (e.g. Necklel et al., 2013; Rankl et al., 2016; Dehecq et al., 2016; Neelmeijer et al., 2017; Vijay et al., 2017; Malz et al., 2018; Abdel Jaber, 2018; Rott et al., 2018). Neelmeijer et al., (2017), provide a clear overview of the processing chain

of TanDEM-X in figure 3 or Dehecq et al, (2016) in figure 2. On the other hand, the TanDEM-X Global DEM is an effort from DLR (German Space Agency) to cover the entire globe with low and high-resolution DEM (12 to 90 m) with thousands of intermediate DEMs to generate this large globe DEM mosaic. Unlike of SRTM DEM (February of 2000), the dates for the composition TanDEM-X "GLOBAL" DEM mosaic is unknown or at least not eas- ily found. Which means for the global DEM, we can find different intermediate DEM seasons. This led to one of the major uncertainties: what are the dates of the all in- termediate DEM in Himalayas? However, although the TanDEM-X "Global" DEM is a very sophisticated DEM that provides a useful topography for many other fields, for glacier elevation changes calculations may lead to more uncertainties. Hence, I leave this point to the discretion of the editor. ##Reply## The consistent concern of uncertainty due to global DEM utility shown by both the reviewers is certainly noted, but it needs to be understood that the global DEM product used is a finished product devoid of much bias as mentioned by DLR with a vertical accuracy up to 2m. Further, to address the issue raised, we have performed the same study over South America as done by Braun et al, 2019 using the TanDEM-X global DEM, and the results are comparable (-0.38 ±0.08 m a-1 for Tierra del Fuego region).Figure attached for reference(Fig. 1). Further, the processing chain for TanDEM-X discussed in references (e.g. Neelmeijer e al., 2017 and Dehecq et al; 2016) are not in the scope of the paper as we utilize finished products, kindly provided by DLR.

2.Comment: I also agree with the reviewer #1 that the present manuscript is missing several previous studies either for the methodology description (e.g. Rankl et al., 2016; Dehecq et al., 2016; Neelmeijer et al., 2017) or for the study area (e.g. King et al., 2017; Brun et al., 2017). The most critical study is from Brun et al., (2017). Brun and colleagues ##Reply## The methodology description has been followed using Gardelle et al.,2013, Vijay and Braun,2016 . Description of study area can be improved as per suggestions of the reviewers.

3.Comment: The description of the methods and uncertainties section, as I stated

before, are not specific enough. I have a lot of doubts about the methodology and the interpretation of results that the authors present in this manuscript, along with the important confusion of the used materials. I will also give you some suggestions, however, substantial work will be needed. ##Reply## Uncertainty analysis of DEM, glacier outline and hypsometry have been performed while carrying out the analysis and incorporated in the total mass balance calculations as per Braun et al., 2019 (Page 5,Line 15). The formula is mentioned in the reference, hence a detailed description of the same was thought to be unnecessary.

4.Methods section:

a)Comment: P2 L17 -> What do you define as rugged terrain? In some areas of the Andes is reaching almost 7000 m a.s.l. ##Reply## Rugged terrains of Himalayas are a well-known characteristic (Pandit et al., 2014; Scherler et al., 2011; Tawde et al., 2017), hence reiterated in the manuscript.

b)Comment: P3 L10-15 -> it is very simplistic to state as 2014, where not further information is provided. As I mentioned above, please try to provide a realistic date to trace the results. I agree with reviewer #1 that the results are biased. ##Reply## Considering the reviewer's suggestions, we found that each DEM tile has a unique set of acquisition dates which encompass ablation as well as accumulation months. For. E.g. Tile N30E79 which falls in Uttarakhand has been generated using 119 scenes both in ascending and descending pass within the time period of Feb 2011- Sept 2014. Hence, for the TanDEM-X DEM product used providing a specific date apart from the end date of acquisition is beyond the scope of this paper.

c)Comment: P4 L3-6 -> how was the radar signal penetration considered? It is not precisely de- scribed. I am not fully convinced with the values showed by the authors. Other examples dealing with X and C band penetration showed much more radar signal penetra- tion (e.g Dehecq et al., (2016); Neelmeijer et al., (2017); Vijay et al., (2017) (see my suggestion below). ##Reply## The values have been compared to reported

literature as that of Gardelle et al., 2013 (Please refer to Table 1).The references cited by the reviewer have a separate study on altitudinal distribution of radar penetration for a limited number of glaciers. As we present a state-wise mean penetration bias, the altitudinal variation might be moderated but not unaccounted for.

d)Comment: P5 L1-3 -> No detailed information about the hypsometry computation and error assessment. ##Reply## Altitudinal distribution/hypsometric computation is a well-known process performed previously (Bhushan et al., 2018; Braun et al., 2019; Vijay and Braun, 2016) without the description of the computation.

e)Comment: P5 L10-19 -> There is information is missing in this section (a) no description for the NMAD method (see Höhle and Höhle, 2009). (b) What equation contains the total uncertainty of your study? (e.g. Vijay et al., 2017; Braun et al., 2019). ##Reply## The uncertainty assessment has been performed as per the description in Braun et al., 2019, which has already been mentioned in the manuscript. The details of NMAD was not mentioned as there are many references which explain the same (Dehecq et al., 2016 and Höhle and Höhle, 2009) However, if the reviewer insists, the relevant references for the readers' benefit shall be added.

f)Comment: Uncertainty from the volume to mass conversion. Please use density scenarios (due to multi-seasonality/dates). ##Reply## Volume to mass conversion uncertainty can be calculated for two different scenarios (850±60 kg/m3 and 900±60 kg/m3) to account for alpine glaciers and glacier ice following Braun et al., 2019. But even with different scenes Brun et al., 2017 seem to have used the well accepted value of 850±60 kg/m3 (please refer P669).

g)Comment: Uncertainty from radar signal penetration. Vijay and Braun (2016) showed that there is a strong altitude dependency of the radar signal penetration bias. They observed a range from 0.84 m (5000 m a.s.l.) to 3.64 m (5800 m a.s.l.). Since the date of season/year of TanDEM-X "Global" DEM is unknown I would use the worst scenario of radar signal penetration. A good example is in Braun et al., (2019), although

they calculated glacier mass changes in the ablation period, they applied a radar signal penetration from 0 to 5 m from Equilibrium Line Altitude (ELA), considering negligible below to ELA. Other example is given by Neelmeijer et al., (2017). You should consider similar procedures. ##Reply## A state wise mean penetration bias has been considered for the glaciated terrain (Gardelle et al., 2013) as we estimate mean mass balance (state-wise). However, the cited references, apart from mentioning the altitudinal dependency, do not provide details of utilizing elevation based radar penetration in their calculation. Further, there is no available physical method to estimate and correct the penetration biases from the radar signal sources. It is just an empirical approach adopted in pertinent literature (Gardelle et al., 2013) .

h)Comment: Uncertainty by the hypsometry (please see Berthier et al., 2016; Brun et al., 2017; Vijay et al., 2016; Braun et al., 2019) i)Error from the DEM differencing (please see Berthier et al., 2016; Vijay et al., 2016; Brun et al., 2017; Braun et al., 2019) j)Error from the glacier outlines (please see Berthier et al., 2016; Brun et al., 2017; Braun et al., 2019) ##Reply## Uncertainty and error analysis of hypsometry, DEM and glacier outlines has been performed as per Braun et al., 2019 and shall be clarified in the manuscript.

k)Comment: Uncertainty by the dates. This point requires investigation/analysis. It would be good if the authors can get some originals intermediate DEMs of TanDEM-X "GLOBAL" to check some dates. ##Reply## Intermediate DEM of the TanDEM-X "global" DEM as per the reviewer's suggestion was pursued. For one tile for example, N32E79_DEM, there are 20 acquisitions from the year 2011, 15 acquisitions from the year 2012, 30 acquisitions in 2013 and 26 acquisitions in 2014. The baselines are varying between 95 to 200 and the scenes such that they cover the entire area each year. So each year the acquisition is made for months varying from February to December, the information in the DEM is updated, hence a mosaic of information provided. Therefore, it if difficult to provide a realistic date. However, to make sure that the authors are not providing wrong information to the readers of this prestigious

journal, we made sure to compare the results with reported literature, finding a good correlation of 0.79. Hence, proving that the result presented in this manuscript are not speculative but rigorously evaluated.

5.Results and discussion section:

a)Comment: First, I would separate results and discussion, since there are some topics you have to properly discuss. e.g. A section on your error assessment with the proposed accuracy assessment methodology. Comparison with other studies and comparison with your glacier mass balance dataset (glaciological method). ##Reply## Authors thank the reviewer for the suggestion but would like to clarify that there is no proposed error assessment methodology. In fact, the errors estimated have been performed as suggested in Braun et al., 2019.

b)Comment: I also suggest you use the catchment/sub-division used by Brun et al., 2017 or Deheq et al., 2018 in order to have comparable numbers in your results and the discussion. ##Reply## The authors thank the reviewer for suggesting a catchment/sub-division study for the current study area, however, the region of study shall partially cover such regions of interest for a comparative study with that of Brun et al., 2017. In Dehecq et al., 2018, only the velocity measurements are provided. Further, for a comparative analysis, only value for Bhutan is available which can be incorporated in the revised manuscript.

c)Comment: In the last few weeks a couple of papers came out with new insight in this region. It would be good to include it (see: Zemp et al., 2019, Wouters et al., 2019; Maurer et al., 2019). ##Reply## We thank the reviewer for brining to our notice these recent literature and shall duly consider the information that can be extracted from them.

6.Figures: a)Comment: P6 Figure 5 -> I agree with reviewer #1. For such a big area I am not sure if this is a representative figure. Please also check Menounos et al., (2019) or Kääb et al., (2012) there are some useful figures that you could apply. ##Reply## For representative purpose, Figure 2 suffices. Figure 5 is just a magnification for some benchmark glaciers on which a lot of study has previously been done and so the readers will be able to relate better.

b)Comment: P7 Figure 2 -> with a better quality of figure 1 you can remove figure 2. ##Reply## This figure is not comparable to Figure 1 such that it can be replaced. Figure 1 shows the study area extent whereas Figure 2 shows the trend of ice-thickness change over the entire region of interest.

c)Comment: P9 and P10 Figure 4 -> some figures are not well represented in the main text. Principally in the discussion. For example, figure 4 d, f, and g. These hypsometric plots present patterns that should be discussed ##Reply## The authors shall elaborate the discussion on Fig 4d, f,g if required. However, the major findings from the hypsometric plots have been sufficiently mentioned in manuscript (please refer P7,L 20-P8,L30)

References:

Bhushan, S., Syed, T. H., Arendt, A. A., Kulkarni, A. V. and Sinha, D.: Assessing controls on mass budget and surface velocity variations of glaciers in Western Himalaya, Scientific Reports, 8(1), 1–11, doi:10.1038/s41598-018-27014-y, 2018.

Braun, A. M. H., Malz, P., Sommer, C. and Barahona, D. F.: Constraining glacier elevation and mass changes in South America, Nature Climate Change letters, 9(region 03), 130–136, doi:10.1038/s41558-018-0375-7, 2019.

Gardelle, J., Berthier, E., Arnaud, Y. and Kääb, A.: Region-wide glacier mass balances over the Pamir-Karakoram-Himalaya during 1999–2011, Cryosphere, 7(4), 1263–1286, doi:10.5194/tc-7-1263-2013, 2013.

Bhushan, S., Syed, T. H., Arendt, A. A., Kulkarni, A. V. and Sinha, D.: Assessing controls on mass budget and surface velocity variations of glaciers in Western Himalaya, Scientific Reports, 8(1), 1–11, doi:10.1038/s41598-018-27014-y, 2018.

Braun, A. M. H., Malz, P., Sommer, C. and Barahona, D. F.: Constraining glacier elevation and mass changes in South America, Nature Climate Change letters, 9(region 03), 130–136, doi:10.1038/s41558-018-0375-7, 2019.

Gardelle, J., Berthier, E., Arnaud, Y. and Kääb, A.: Region-wide glacier mass balances over the Pamir-Karakoram-Himalaya during 1999–2011, Cryosphere, 7(4), 1263–1286, doi:10.5194/tc-7-1263-2013, 2013.

Kääb, A., Berthier, E., Nuth, C., Gardelle, J. and Arnaud, Y.: Contrasting patterns of early twenty-first-century glacier mass change in the Himalayas, Nature, 488(7412), 495–498, doi:10.1038/nature11324, 2012.

Manickam, S., Bhattacharya, A., Singh, G. and Yamaguchi, Y.: Estimation of Snow Surface Dielectric Constant from Polarimetric SAR Data, IEEE Journal of Selected Topics in Applied Earth Observations and Remote Sensing, 10(1), 211–218, doi:10.1109/JSTARS.2016.2588531, 2017.

Pandit, A., Ramsankaran, R. and Rao, Y. S.: Generation and Validation of the Interferometric SAR DEMs from TanDEM-X data for Gangotri and Hamtah Glaciers of Indian Himalayas, Procedia Technology, 16, 793–805, doi:10.1016/j.protcy.2014.10.029, 2014.

Scherler, D., Bookhagen, B. and Strecker, M. R.: Spatially variable response of Himalayan glaciers to climate change affected by debris cover, Nature Geoscience, 4(3), 156–159, doi:10.1038/ngeo1068, 2011.

Singh, G., Verma, A., Kumar, S., Snehmani, Ganju, A., Yamaguchi, Y. and Kulkarni, A. V.: Snowpack Density Retrieval Using Fully Polarimetric TerraSAR-X Data in the Himalayas, IEEE Transactions on Geoscience and Remote Sensing, 55(11), 6320–6329, doi:10.1109/TGRS.2017.2725979, 2017.

Tawde, S. A., Kulkarni, A. V. and Bala, G.: An estimate of glacier mass balance for the Chandra basin, western Himalaya, for the period 1984-2012, Annals of Glaciology,

58(75), 99–109, doi:10.1017/aog.2017.18, 2017.

Vijay, S. and Braun, M.: Elevation change rates of glaciers in the Lahaul-Spiti (Western Himalaya, India) during 2000-2012 and 2012-2013, Remote Sensing, 8(12), 1–16, doi:10.3390/rs8121038, 2016.

[Figure]

**Fig. 1.**